# GUIDING REPRESENTATION LEARNING IN DEEP GENERATIVE MODELS WITH POLICY GRADIENTS

## ABSTRACT

Variational Auto Encoder (VAE) provide an efficient latent space representation of complex data distributions which is learned in an unsupervised fashion. Using such a representation as input to Reinforcement Learning (RL) approaches may reduce learning time, enable domain transfer or improve interpretability of the model. However, current state-of-the-art approaches that combine VAE with RL fail at learning good performing policies on certain RL domains. Typically, the VAE is pre-trained in isolation and may omit the embedding of task-relevant features due to insufficiencies of its loss. As a result, the RL approach can not successfully maximize the reward on these domains. Therefore, this paper investigates the issues of joint training approaches and explores incorporation of policy gradients from RL into the VAE's latent space to find a task-specific latent space representation. We show that using pre-trained representations can lead to policies being unable to learn any rewarding behaviour in these environments. Subsequently, we introduce two types of models which overcome this deficiency by using policy gradients to learn the representation. Thereby the models are able to embed features into its representation that are crucial for performance on the RL task but would not have been learned with previous methods.

## 1 INTRODCTION

Reinforcement Learning (RL) gained much popularity in recent years by outperforming humans in games such as *Atari* (Mnih et al. (2015)), *Go* (Silver et al. (2016)) and *Starcraft 2* (Vinyals et al. (2017)). These results were facilitated by combining novel machine learning techniques such as deep neural networks (LeCun et al. (2015)) with classical RL methods. The RL framework has shown to be quite flexible and has been applied successfully in many further domains, for example, robotics (Andrychowicz et al. (2020)), resource management (Mao et al. (2016)) or physiologically accurate locomotion (Kidziński et al. (2018)).

The goal of representation learning is to learn a suitable representation for a given application domain. Such a representation should contain useful information for a particular downstream task and capture the distribution of explanatory factors (Bengio et al. (2013)). Typically, the choice of a downstream task influences the choice of method for representation learning. While Generative Adversarial Network (GAN)s are frequently used for tasks that require high-fidelity reconstructions or generation of realistic new data, auto-encoder based methods have been more common in RL. Recently, many such approaches employed the Variational Auto Encoder (VAE) (Kingma & Welling (2013)) framework which aims to learn a smooth representation of its domain.

Most of these approaches follow the same pattern: First, they build a dataset of states from the RL environment. Second, they train the VAE on this static dataset and lastly train the RL mode using the VAE's representation. While this procedure generates sufficiently good results for certain scenarios, there are some fundamental issues with this method. Such an approach assumes that it is possible to collect enough data and observe all task-relevant states in the environment without knowing how to act in it. As a consequence, when learning to act the agent will only have access to a representation that is optimized for the known and visited states. As soon as the agent becomes more competent, it might experience novel states that have not been visited before and for which there is no good representation (in the sense that the experienced states are out of the original learned distribution and the mapping is not appropriate).

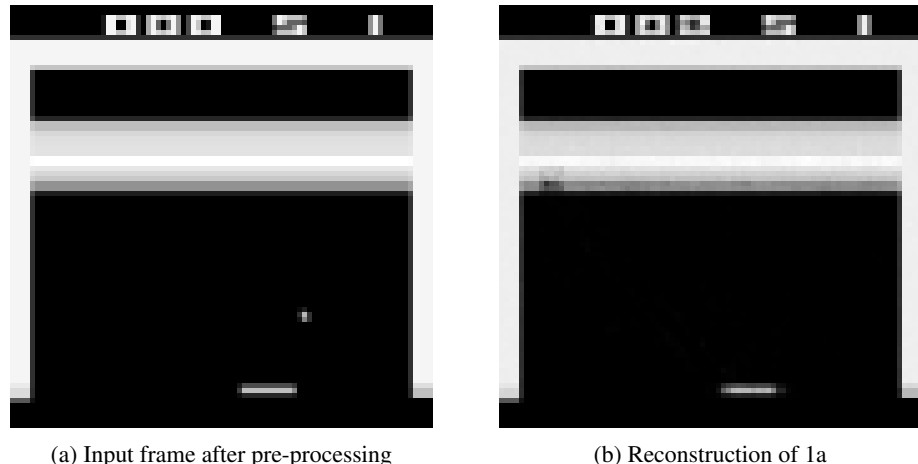

(a) Input frame after pre-processing      (b) Reconstruction of 1a

Figure 1: A frame from Atari Breakout. The original image 1a was passed through a pre-trained VAE yielding the reconstruction 1b. Note the missing ball in the lower right hand corner.

Another issue arises from the manner the representation is learned. Usually, the VAE is trained in isolation, so it decides what features are learned based on its own objective function and not on what is helpful for the downstream task. Mostly, such a model is tuned for good reconstruction. Without the information from the RL model, such a representation does not reflect what is important for the downstream task. As a consequence, the VAE might omit learning features that are crucial for good performance on the task because they appear negligible with respect to reconstruction (Goodfellow et al. (2016), Chapter 15, Figure 15.5). For example, small objects in pixel-space are ignored as they affect a reconstruction based loss only marginally. Thus, any downstream task using such a representation will have no access to information about such objects. A good example for such a task is Atari Breakout, a common RL benchmark. Figures 1a and 1b show an original Breakout frame and its reconstruction. While the original frame contains the ball in the lower right hand corner, this crucial feature is missing completely in the reconstruction.

We approach this issue through simultaneously learning representation and RL task, that is by combining the training of both models. As an advantage, this abolishes the need of collecting data before knowing the environment as it combines VAE and RL objectives. In consequence the VAE has an incentive to represent features that are relevant to the RL model. The main contributions of this paper are as follows: First we show that combined learning is possible and that it yields good performing policies. Second, we show that jointly trained representations incorporate additional, task-specific information which allows a RL agent to achieve higher rewards then if it was trained on a static representation. This will be shown indirectly by comparing achieved rewards as well as directly through an analysis of the trained model and its representation.

## 2 RELATED WORK

Lange & Riedmiller (2010) explored Auto Encoder (AE) (Lecun (1987); Bourlard & Kamp (1988); Hinton & Zemel (1994)) as a possible pre-processor for RL algorithms. The main focus in their work was finding good representations for high dimensional state spaces that enables policy learning. As input, rendered images from the commonly used grid world environment were used. The agent had to manoeuvre through a discretized map using one of four discrete movement actions per timestep. It received a positive reward once reaching the goal tile and negative rewards elsewhere. The AE bottleneck consisted only of two neurons, which corresponds to the dimensionality of the environemnt's state. Fitted Q-Iteration (FQI) (Ernst et al. (2005)) was used to estimate the $Q$-function, which the agent then acted $\epsilon$-greedy upon. Besides RL, they also used the learned representation to classify the agents position given an encoding using a Multi-Layer Perceptron (MLP) (Rumelhart et al. (1985)). For these experiments, they found that adapting the encoder using MLP

gradients lead to an accuracy of $99.46\%$. However, they did not apply this approach to their RL task.

A compelling example for separate training of meaningful representation is provided by Higgins et al. (2017b) who proposed a framework called $DARLA$. They trained RL agents on the encoding of a $\beta$-VAE (Higgins et al. (2016); Higgins et al. (2017a)) with the goal of zero-shot domain transfer. In their approach, $\beta$-VAE and agent were trained separately on a source domain and then evaluated in a target domain. Importantly, source and target domain are similar to a certain extent and only differ in some features, e.g. a blue object in the source domain might be red in the target domain. During training of the $\beta$-VAE, the pixel-based reconstruction loss was replaced with a loss calculated in the latent space of a Denoising Auto Encoder (DAE) (Vincent et al. (2008)). Thereby their approach avoids missing task relevant feature encodings at the cost of training another model. For one of their evaluation models, they allowed the RL gradients to adapt the encoder. Their results show that subsequent encoder learning improves performance of Deep Q-Learning (DQN) but decreases performance of Asynchronous Advantage Actor-Critic (A3C) (Mnih et al. (2016)).

Ha & Schmidhuber (2018) proposed a combination of VAE, Recurrent Neural Networks (RNN) (Hochreiter & Schmidhuber (1997)) and a simple policy as a controller. They hypothesized that by learning a good representation of the environment and having the ability to predict future states, learning the policy itself becomes a trivial task. Like in most other models, the VAE was pre-trained on data collected by a random policy. Only the RNN and the controller were trained online. The compressed representation from the VAE was passed into a RNN in order to estimate a probability density for the subsequent state. The controller was deliberately chosen as a single linear layer and could thus be optimized with Covariance Matrix Adaptation - Evolution Strategy (CMA-ES) (Hansen (2006)).

This work demonstrated how a VAE can provide a versatile representation that can be utilized in reinforcement learning. In addition, such an approach allows to predict the subsequent encoded state. While these findings encourage the usage of VAE in conjunction with RL, this is only possible in environments where the state space can be explored sufficiently by a random policy. However, if the policy can only discover important features after acquiring a minimal level of skill, sampling the state space using a random policy will not yield high-performing agents. Learning such features would only be possible if the VAE is continuously improved during policy training.

Another interesting combination of VAEs and RL was recently proposed by Yang et al. (2019), with their so called Action-Conditional Variational Auto-Encoder (AC-VAE). Their motivation for creating this model was to train a transparent, interpretable policy network. Usually, the $\beta$-VAEs decoder is trained to reconstruct the input based on the representation the encoder produced. In this work though, the decoders objective was to predict the subsequent state $s_{t+1}$. As input it got the latent space vector $z$ combined with an action-mapping-vector, which is the action vector $a_t$ with a zero-padding to match the latent spaces dimensionality. Inspecting the decoder estimates for $s_{t+1}$ when varying one dimension of the latent space showed, that each dimension encoded a possible subsequent state that is likely to be encountered if the corresponding action from this dimension was taken. Unfortunately, the authors did not report any rewards they achieved on Breakout, hence it was not possible for us to compare model performances.

## 3  COMBINATION OF REINFORCEMENT AND REPRESENTATION LEARNING OBJECTIVES

In this section, we will first revisit the fundamentals of RL and VAEs and discuss their different objective functions. Then, we propose a joint objective function that allows for joint training of both models using gradient descent based learning methods.

### 3.1  REINFORCEMENT LEARNING WITH POLICY OPTIMIZATION

RL tries to optimize a Markov Decision Process (MDP) (Bellman (1957)) that is given by the tuple $\langle \mathcal{S}, \mathcal{A}, r, p, \gamma \rangle$. $\mathcal{S}$ denotes the state space, $\mathcal{A}$ the action space and $p : \mathcal{S} \times \mathcal{R} \times \mathcal{S} \times \mathcal{A} \rightarrow [0, 1]$ the environment's dynamics function that, provided a state-action pair, gives the state distribution for the successor state. $r : \mathcal{S} \times \mathcal{A} \rightarrow \mathcal{R}$ is the reward and $\gamma \in [0, 1)$ the scalar discount factor. The policy $\pi_\theta(a|s)$ is a stochastic function that gives a probability distribution over actions for state

$s$. $\theta$ denotes the policy's parameter vector which is typically subject to optimization. A trajectory $\tau = (s_0, a_0, ..., s_T, a_T)$ consisting of an alternating sequence of states and actions can be sampled in the environment, where $T$ stands for the final timestep of the trajectory and $a_i \sim \pi_\theta(a_i|s_i)$.

The overarching goal of RL is to find a policy that maximizes the average collected reward over all trajectories. This can be expressed as the optimization problem $\max \mathbb{E}_{\tau \sim p(\tau)} \left[ \sum_t r(s, a) \right]$, which can also be written in terms of an optimal policy parameter vector $\theta^* = \arg \max_\theta \mathbb{E}_{\tau \sim p(\tau)} \left[ \sum_t r(s, a) \right]$. When trying to optimize the policy directly be searching for $\theta^*$, policy optimization algorithms like A3C, Actor-Critic with Experience Replay (ACER) (Wang et al. (2016a)), Trust Region Policy Optimization (TRPO) (Schulman et al. (2015a)) or Proximal Policy Optimization (PPO) (Schulman et al. (2017)) are commonly used. The fundamental idea behind policy optimization techniques is to calculate gradients of the RL objective with respect to the policy parameters:

$$\nabla_\theta J(\theta) = \mathbb{E}_{\tau \sim p(\tau)} \left[ \nabla_\theta \log \pi_\theta(\tau) \, r(\tau) \right] \tag{1}$$

where we defined $\sum_{t=0}^{T} r(s, a) = r(\tau)$ for brevity. However, most policy optimization methods introduce heavy modifications to this vanilla gradient in order to achieve more stable policy updates. Throughout our work, we have used PPO as RL algorithm because it is quite sample efficient and usually produces stable policy updates. For an in-depth description of PPO, we refer to our A.1 or the original work Schulman et al. (2017).

## 3.2 Learning Representations using Variational Auto-Encoders

Kingma & Welling (2013) introduced the VAE as a method to perform Variational Inference (VI) (Jordan et al. (1999)) using function approximators, e.g. deep neural networks. VI tries to approximate a distribution over the generative factors of a dataset which would otherwise involve calculating an intractable integral. The authors present an algorithm that utilizes the auto encoder framework, an unsupervised learning method which learns data encodings by reconstructing its input. Therefore, the input is first compressed until it reaches a given size and is afterwards decompressed to its original size. When using deep neural networks, these transformations can be achieved by using for example fully connected or convolutional layers. In order for the VAE to approximate a distribution over generative factors, the authors used the so called "reparametrization trick". It allows for gradient based optimization methods to be used in searching for the distribution parameters. For training the VAE, a gradient based optimizer tries to minimize the following loss:

$$\mathcal{L}^{VAE}(\boldsymbol{x}, \phi, \psi) = -D_{KL}(q_\phi(\boldsymbol{z}|\boldsymbol{x}) \, || \, p(\boldsymbol{z})) + \mathbb{E}_{q_\phi(\boldsymbol{z}|\boldsymbol{x})} \left[ \log p_\psi(\boldsymbol{x}|\boldsymbol{z}) \right] \tag{2}$$
$$\text{with } \boldsymbol{z} = l(\boldsymbol{\mu}, \boldsymbol{\sigma}, \boldsymbol{\epsilon}) \text{ and } \boldsymbol{\epsilon} \sim p(\boldsymbol{\epsilon})$$

where $D_{KL}$ denotes the Kullback-Leibler Divergence (KL) (Kullback & Leibler (1951)) of the approximated distribution over generative factors produced by the encoder $q_\phi(\boldsymbol{z}|\boldsymbol{x})$ and some prior distribution $p(\boldsymbol{z})$. The expectation is often referred to as reconstruction loss that is typically calculated on a per-pixel basis. Lastly, $l(\boldsymbol{\mu}, \boldsymbol{\sigma}, \boldsymbol{\epsilon})$ is a sampling function that is differentiable w.r.t. the distribution parameters, for example $\boldsymbol{z} = \boldsymbol{u} + \boldsymbol{\sigma}\boldsymbol{\epsilon}$.

## 3.3 Joint Objective Function

Combining both loss functions such that both models can be trained at the same time is rather straight-forward. Adding both individual losses and using an optimizer such as ADAM (Kingma & Ba (2014)) to minimize them is sufficient to achieve joint training. During backpropagation, gradients from the policy and the VAE are combined in the latent space. Due to different topologies of the networks, gradient magnitudes differ significantly. Therefore, we introduced the hyperparameter $\kappa$ which can be used to either amplify or dampen the gradients and we arrive at the following loss:

$$\mathcal{L}^{\text{joint}} = \kappa \mathcal{L}^{\text{PG}}(\theta_k, \theta_{k-1}, \phi_k) + \mathcal{L}^{VAE}(\boldsymbol{x}, \phi, \psi, \beta) \tag{3}$$

where $\mathcal{L}^{\text{PG}}$ is some policy gradient algorithm's objective function. As mentioned before, we used PPO's loss $\mathcal{L}^{\text{PPO}}$ (equation 4 in the appendix).

## 4 EXPERIMENTS

In order to test our model with the combined objective function given by equation 3, we have used the well-known benchmark of Atari Breakout. This environment has several properties that make it appealing to use: it is easily understandable by humans, used often as a RL task and the conventional pre-trained methods fail at mastering it. The ball is the most important feature that is required to be encoded in order to perform well, is heavily underrepresented (approximately 0.1% of the observation space). Therefore, the VAE's incentive to encode it is very low whereas our model succeeds in encoding it. In the following, we compare the pre-trained approach to two different continuously trained models that use the loss from equation 3.

### 4.1 DATA COLLECTION AND PRE-PROCESSING

The raw RGB image data produced by the environment has a dimensionality of $210 \times 160 \times 3$ pixels. We employ a similar pre-precessing as Mnih et al. (2015), but instead of cropping the grey-scaled frames, we simply resize them to $84 \times 84$ pixels. As we will first train models similar to those introduced in previous works with a pre-trained VAE, we needed to construct a dataset containing Breakout states. We used an already trained policy to collect a total of $25,000$ frames, the approximate equivalent of 50 episodes.

### 4.2 PRE-TRAINING THE VARIATIONAL AUTO-ENCODER

Our first model is based on those of the previously introduced works which involve isolated pre-training the VAE on a static dataset. Figure 2 shows the individual parts of the complete training process. For the first model, PPO[fixed], the encoder and decoder (shown in orange and red) are pre-trained before policy training. During this phase, there is no influence from the RL loss. Once the VAE training is finished, the decoder shown in red in Figure 2 is discarded completely. Later during

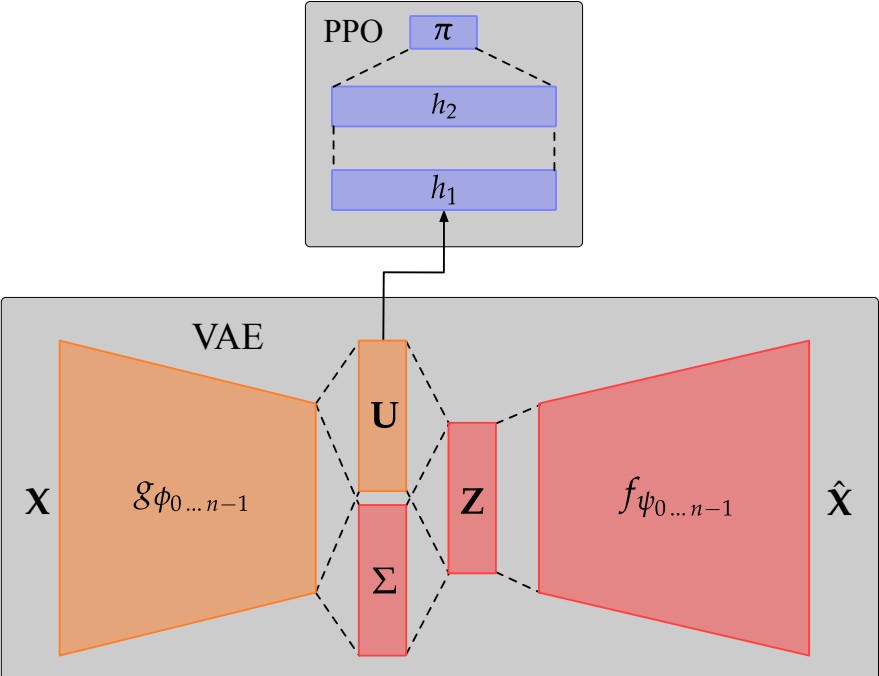

Figure 2: Model combining PPO and a VAE. Depending on the model configuration, the colored parts are trained differently. $\mathbf{X}$ is the VAE's input and $\hat{\mathbf{X}}$ the reconstructions. PPO receives the mean vectors $\mathbf{U}$ as input and calculates a distribution over actions $\pi$. Note that we use capital letters in the VAE to emphasize that we pass $n$ frames at the same time when a policy is trained.

policy training, we use $n$ instances of the same encoder with shared weights that receive a sequence of the last $n$ frames as input. Stacking allows us to incorporate temporal information and for the policy to predict the ball's trajectory. By sharing the weights, we ensure that the resulting encodings originate from the same function. $\mathbf{U}$ then represents the concatenated encodings of the sequence. This weight sharing method has proven to be more time efficient than to query the encoder $n$ times and concatenate the results afterwards.

Prior to policy training, we trained the VAE on the dataset we have collected before, with hyperparameters from table 1. Once pre-training was finished, we discarded the decoder weights and used the stacked encoder as input for the policy MLP. The MLP was then trained 10M steps with hyperparameters from table 2. During this training, the encoder weights were not changed by gradient updates anymore but remained fixed.

The second model we introduce is called PPO$^{\text{adapt}}$, which has the same structure and hyperparameters as the first model. For this model, we also train the VAE in isolation first, however the encoder weights are not fixed anymore during policy training. Gradients from the RL objective are back propagated through the encoder, allowing it to learn throughout policy training. We hypothesize that features that are important for policy performance can be incorporated in an already learned representation.

Figure 3 compares the median rewards of three rollouts with different random seeds for all models. PPO$^{\text{fixed}}$ was not once able to achieve a reward of 10 or higher, while PPO$^{\text{adapt}}$ steadily improved its performance with final rewards well over 50. The learning curve of PPO$^{\text{adapt}}$ shows that the model is able to learn how to act in the environment, whereas PPO$^{\text{fixed}}$ does not. The non-zero rewards from PPO$^{\text{fixed}}$ are similar to those of random agents in Breakout. From these results, we can assume that training the VAE in isolation on a static dataset for Breakout results in a deficient representation for RL. Therefore, using policy gradients to adapt an already learned representation can be beneficial in environments where the VAE fails to encode task-relevant features.

### 4.3 Jointly Learning Representation and Policy

The last model we introduce, PPO$^{\text{VAE}}$, combines a complete VAE with a policy MLP that receives $\mathbf{U}$, the concatenated state encodings, as input. As opposed to the first two models, all weights are initialized randomly before policy training and the VAE is not pre-trained. For this procedure an

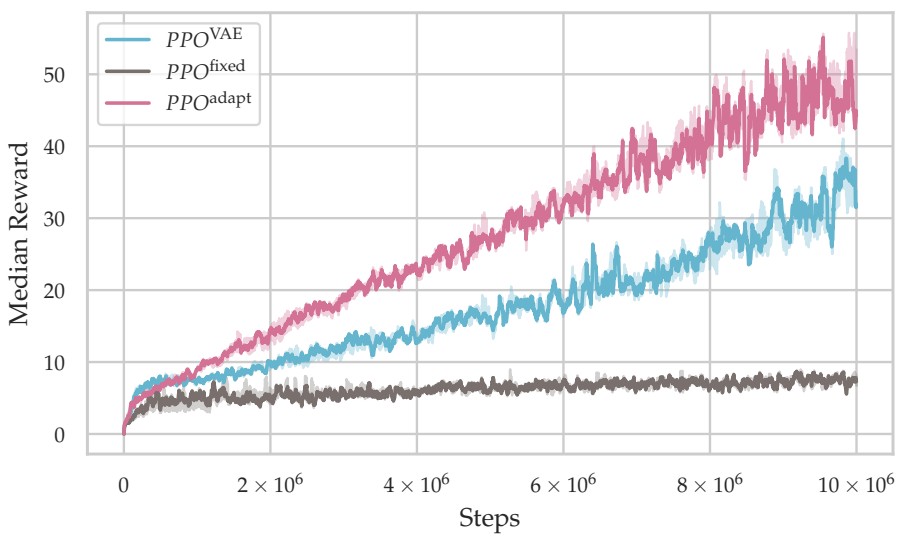

Figure 3: Reward of the three proposed models across three random seeds each. PPO$^{\text{fixed}}$ is not able to achieve high rewards while the other two models consistently improve their performance.

already trained agent that gathers a dataset for the VAE beforehand is not necessary. The decoder is trained exactly as in the isolated setting, meaning its gradients are also only computed using the VAE's loss function. During backpropagation, the gradients coming from $\mathbf{Z}$ and $h_1$ are added together and passed through the encoder. This model has the same network configuration and hyperparameters as the first two, with the only difference that we also evaluated different values for $\kappa$ from the joint loss equation 3 (see A.3). For the results reported here, we chose $\kappa = 20$. All hyperparameters can be found in table 3.

By simultaneously training representation and policy, we expect the VAE to learn task-relevant features from the beginning of training. This assumption is supported by the learning curve shown in figure 3, which compares PPO$^{\text{VAE}}$ to the previous two models. The curve shows a steady increase in reward over the course of training with PPO$^{\text{VAE}}$ achieving slightly higher rewards than PPO$^{\text{adapt}}$ in the beginning. This characteristic changes after less than 1M steps and from that point on PPO$^{\text{adapt}}$ consistently outperforms PPO$^{\text{VAE}}$. This difference in performance is likely attributed to the fact, that in PPO$^{\text{VAE}}$ the decoder is trained throughout the complete training. While for PPO$^{\text{adapt}}$, the latent space can be changed without restrictions, the decoder of PPO$^{\text{VAE}}$ constantly produces gradients do not contain information about the ball. Therefore, PPO$^{\text{VAE}}$ has less incentive to achieve higher rewards, which is reflected by its performance.

### 4.4 Analyzing the Value Function Gradients

So far, the results imply that PPO$^{\text{VAE}}$ and PPO$^{\text{adapt}}$ do indeed learn encodings of the ball. One difficulty when analyzing the representation is, that the decoder still has no incentive to reconstruct the ball, even if it is present in the latent space. In a work that enhances DQN algorithm (Wang et al. (2016b)), the authors visualized the Jacobian of the value function w.r.t. the input images. These visualizations showed which features or regions from the input space are considered as important in terms of future reward. As we also learn a value function, we did the same and visualized what our model considered important and what not.

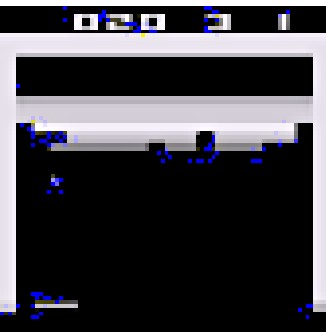

Figure 4: The Jacobian of PPO's value function. Highlighted areas mean high importance in terms of future rewards. Note the high Jacobian values around the ball and the blocks.

In figure 4 we illustrate a pre-processed frame and added the values of the Jacobian to the blue channel if the were greater than the mean value of the Jacobian. Only visualizing above-mean Jacobian values removes some noise in the blue channel makes the images much easier to interpret and only highlights regions of high relevance. We can clearly see, that the Jacobian has high values at missing blocks as well as around the ball, meaning that these regions are considered to have high impact on future rewards. By visualizing the Jacobian we have confirmed that the policy gradients encourage the VAE to embed task-relevant features.

## 5 Conclusion

This paper focused on the issue of pre-training VAEs with the purpose of learning a policy for a downstream task based on the VAE's representation. In many environments, the VAE has little to no incentive to learn task-relevant features if they are small in observation space. Another issue arises if the observation of these features depends on policy performance and as a result, they are underrepresented in a dataset sampled by a random agent. In both cases, fixing encoder weights during policy training prevents the VAE to learn these important features and policy performance will be underwhelming.

We carried out experiments on the popular RL benchmark Atari Breakout. The goal was to analyze whether policy gradients guide representation learning towards incorporating performance-critic features that a VAE would not learn on a pre-recorded dataset. First experiments confirmed, that the common pre-trained approach did not yield well-performing policies in this environment. Allowing the policy gradients to adapt encoder weights in two different models showed significant

improvements in terms of rewards. With policy gradients guiding the learned representation, agents consistently outperformed those that were trained on a fixed representation.

Our work verifies that the fundamental issue with pre-trained representations still exists and shows possible solutions in RL scenarios. Nonetheless, future work can still explore a variety of improvements to our models. For once, training not only the encoder but also the decoder with RL gradients can improve interpretability of the VAE and enable it to be used as a generator again that also generates task-relevant features. Another direction is to impose further restrictions on the latent space during joint training of VAE and policy. The goal there would be to maintain the desired latent space characteristics of VAEs while still encoding task-relevant features.

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

## A    APPENDIX

### A.1    STABLE POLICY LEARNING WITH PROXIMAL POLICY OPTIMIZATION

Most actor-critic algorithms successfully reduce the variance of the policy gradient, however they show high variance in policy performance during learning and are at the same time very sample inefficient. Natural gradient (Amari (1998)) methods such as TRPO from Schulman et al. (2015a) greatly increase sample efficiency and learning robustness. Unfortunately, they are relatively complicated to implement and are computationally expensive as the require some second order approximations. PPO (Schulman et al. (2017)) is a family of policy gradient methods that form pessimistic estimates of the policy performance. By clipping and therefore restricting the policy updates, PPO prohibits too large of a policy change as they have been found to be harmful to policy performance in practice. PPO is often combined with another type of advantage estimation (Schulman et al. (2015b)) that produces high accuracy advantage function estimates.

We define the PPO-Clip objective is defined as

$$J^{\mathrm{PPO}}(\theta_k, \theta_{k-1}) = \mathbb{E}\left[\min\left(o(\theta)A^{\pi_{\theta_k}}(s,a), \mathrm{clip}\left(o(\theta), 1-\epsilon, 1+\epsilon\right)A^{\pi_{\theta_k}}(s,a)\right)\right]$$

$$\mathrm{s.t.}\ \ \delta_{\mathrm{MB}} < \delta_{\mathrm{target}}$$

(4)

where $o(\theta) = \frac{\pi_{\theta_k}(a|s)}{\pi_{\theta_{k-1}}(a|s)}$ denotes the probability ratio of two policies.

This objective is motivated by the hard KL constraint that TRPO enforces on policy updates. Should a policy update result in a policy that deviates too much from its predecessor, TRPO performs a line search along the policy gradient direction that decreases the gradient magnitude. If the constraint is satisfied during the line search, the policy is updated using that smaller gradient step. Otherwise the update is rejected after a certain number of steps. This method requires to calculate the second order derivative of the KL divergence, which is computationally costly. PPO uses its clipping objective to implicitly constrain the deviation of consecutive policies. In some settings, PPO still suffers from diverging policy updates (OpenAI Spinning Up - PPO (2018)), so we included a hard KL constrained on policy updates. The constraint can be checked after each mini-batch update analytically and is therefore not very computationally demanding.

PPO extends the policy gradient objective function from Kakade (2002). With the probability ratio $o(\theta)$, we utilize importance sampling in order to use samples collected with any policy to update our current one. Thereby we can use samples more often than in other algorithms, making PPO more sample efficient. Using importance sampling, we still have a correct gradient estimate. Combining the new objective with actor-critic methods yields algorithm 1. $K$ denotes the number of optimization epoch per set of trajectories and $B$ denotes the mini-batch size. In the original paper, a combined objective function is also given with:

$$\mathcal{L}^{\mathrm{PPO}}(\theta_k, \theta_{k-1}, \phi_k) = \mathbb{E}\left[c_1 J^{\mathrm{PPO}}(\theta_k, \theta_{k-1}) - c_2 \mathcal{L}^{V^{\pi_\theta}}(\phi_k) + \mathcal{H}(\pi_{\theta_k})\right]$$

$$\mathrm{s.t.}\ \ \delta_{\mathrm{MB}} < \delta_{\mathrm{target}}$$

(5)

where $\mathcal{H}(\pi_{\theta_k})$ denotes the policy entropy. Encouraging the policy entropy not to decrease too much prohibits the policy from specializing on one action. As discussed in OpenAI Spinning Up - PPO (2018), there are two cases for $J^{\mathrm{PPO}}(\theta_k, \theta)$: either the advantage function was positive or negative. In case the advantage is positive, it can be written as:

$$J^{\mathrm{PPO}}(\theta_k, \theta) = \mathbb{E}\left[\min\left(o(\theta), (1+\epsilon)\right)A^{\pi_{\theta_k}}(s,a)\right]$$

(6)

$A^{\pi_{\theta_k}}(s,a) > 0$ indicates that the action yields higher reward than other actions in this state, hence we want its probability $\pi_{\theta_k}(a|s)$ to increase. This increase is clipped to $(1+\epsilon)$ once $\pi_{\theta_k}(a|s) > \pi_{\theta_{k-1}}(a|s)(1+\epsilon)$. Note however, that updates that would worsen policy performance are neither clipped nor bound. If the the advantage is negative, it can be expressed as:

$$J^{\mathrm{PPO}}(\theta_k, \theta) = \mathbb{E}\left[\max\left(o(\theta), (1-\epsilon)\right)A^{\pi_{\theta_k}}(s,a)\right]$$

(7)

This equation behaves conversely to 6: $A^{\pi_{\theta_k}}(s,a) < 0$ indicates that we chose a suboptimal action, thus we want to decrease its probability. Once $\pi_{\theta_k}(a|s) < \pi_{\theta_{k-1}}(a|s)(1-\epsilon)$, the max bounds the magnitude by which the action's probability can be decreased.

---

**Algorithm 1** Proximal Policy Optimisation with KL constraint

---

1: Initialize policy parameters $\theta_0$ and value function parameters $\phi_0$
2: **for** $k = 0, 1, 2, ...$ **do**
3:     Collect set of trajectories $\mathcal{D}_k = \{\tau_i\}$ with $\pi_{\theta_k}$ and compute $\hat{R}_t$
4:     $\delta_{\mathrm{MB}} \leftarrow 0$
5:     **for** $0, 1, 2, ...K$ **do**
6:         **for each** mini-batch of size $B$ in $\{\tau_i\}$ **do**
7:             Update the policy by maximizing the PPO-Clip objective 4
8:             Minimize $\mathcal{L}^{V^{\pi_\theta}}$ on the mini-batch
9:         **end for**
10:     **end for**
11:     **if** $\delta_{\mathrm{MB}} > \delta_{\mathrm{target}}$ **then**
12:         $\theta_{k+1} = \theta_k$
13:     **end if**
14: **end for**

---

## A.2   Hyperparameter Tables

| Parameter | Value |
|---|---|
| epochs | 100 |
| batch size | 128 |
| input size | $(84, 84, 1)$ |
| optimizer | ADAM |
| learning rate | $1 \times 10^{-4}$ |
| encoder | Conv2D  $32 \times 4 \times 4$ (stride 2) - $64 \times 4 \times 4$ (stride 2) - FC 512 (ReLU) |
| latents | 20 (linear) |
| decoder | FC 512 (ReLU) - $64 \times 4 \times 4$ (stride 2) - $32 \times 4 \times 4$ (stride 2) Conv2D Transpose |

Table 1: Hyperparameter table for VAE training on Breakout

| Parameter | Value |
|---|---|
| timesteps | $1 \times 10^7$ |
| environments | 16 |
| batch size | 32 |
| $t_{\max}$ | 2048 |
| $K$ | 10 |
| $c_1$ | 1.0 |
| $c_2$ | 0.5 |
| $c_3$ | 0.0 |
| $\gamma$ | 0.99 |
| $\lambda$ | 0.95 |
| network | FC 64 (tanh) - FC 64 (tanh) |
| optimizer | ADAM |
| learning rate | $3 \times 10^{-4}$ |

Table 2: Policy hyperparameters of PPO$^{\mathrm{fixed}}$ and PPO$^{\mathrm{adapt}}$

| Parameter | Value |
|---|---|
| timesteps | $1 \times 10^7$ |
| environments | 16 |
| batch size | 32 |
| $t_{\max}$ | 2048 |
| $K$ | 10 |
| $c_1$ | 1.0 |
| $c_2$ | 0.5 |
| $c_3$ | 0.0 |
| $\gamma$ | 0.99 |
| $\lambda$ | 0.95 |
| network | FC 64 (tanh) - FC 64 (tanh) |
| optimizer | ADAM |
| learning rate | $3 \times 10^{-4}$ |
| $\kappa$ | $(1, 10, 20)$ |

Table 3: Policy hyperparameter table of PPO$^{\mathrm{VAE}}$

## A.3 CHOOSING APPROPRIATE VALUES FOR $\kappa$

In equation 3, we introduced the hyperparameter $\kappa$ to balance VAE and PPO gradients. We found empirically, that tuning $\kappa$ is straight forward and requires only few trials. In order to simplify the search for $\kappa$, one can evaluate gradient magnitudes of the different losses at the point where they are merged at $\mathbf{U}$. Our experiments showed PPO's gradients to be significantly smaller, thus scaling up the loss function was appropriate. This will likely differ if the networks are configured differently. Increasing $\kappa$ from 1 to 10 led to considerably higher rewards, however the difference in performance was small when increasing $\kappa$ further to 20. Therefore, we chose $\kappa = 20$ in our reported model performances.

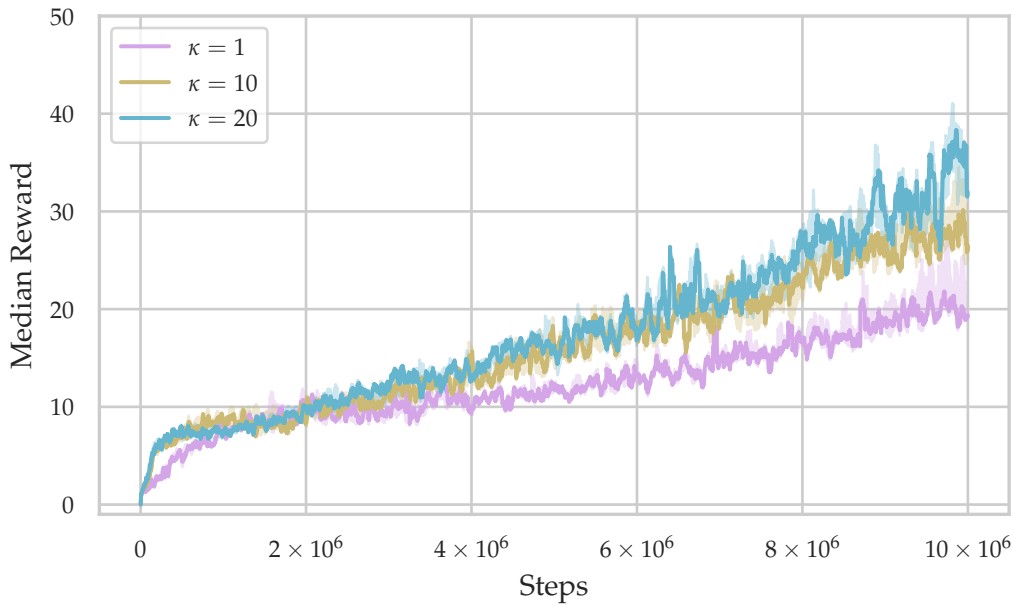

Figure 5: Performance comparison of PPO$^{\text{VAE}}$ with different values for $\kappa$

