# OpenReview forum: "Guiding Representation Learning in Deep Generative Models with Policy Gradients"
_ICLR.cc/2021/Conference — Reject_

### Official Review · AnonReviewer3 · 2020-10-27
**In this work the authors tackle the problem of learning good, task-relevant representations for Reinforcement Learning (RL) with policy gradient (PG) methods. Specifically they propose combining the loss of a variational auto-encoder (VAE) with the PG loss during training to learn better features. They compare this approach against a pre trained VAE model with and without fine-tuning on the Atari Breakout task.**

**Rating:** 2
**Confidence:** 4

**Review:**

My recommendation would be to reject this paper for reasons which I will outline below.

Pros:
The problem of learning good features for RL is an important one to tackle and has application in many important tasks with complex input spaces like vision. The authors explain why this is important and I encourage them to continue to explore this line of research albeit more thoroughly as detailed below.

Cons:
One big issue with the paper is in its lack of experimental rigour. This can be improved by considering more than just the single Breakout task (the Atari suite itself contains many example tasks ripe for research) and also by comparing against more competitive baselines.
For instance the authors mention the AC-VAE (Yang et al. [1]) but do not compare against their method since ‘they do not report rewards achieved on Breakout’. To me this is not a good enough reason to not have a comparison. Instead I encourage the authors to reimplement the alternative approach and compare it on the tasks they consider.
While the paper lists some early work on representation learning and learning features for RL they do not mention or compare against many other works (e.g. Jaderberg et. al. [2] and Achiam et al [3] to name a few). A thorough literature survey helps place the work in the broader context of the field and allows the work to be evaluated more comprehensively.

Finally there are a number of minor points in the presentation which did not affect my final decision but would be good to fix.  There is a typo where 'Introduction' is spelled 'Introdction'. Reference capitalisation is inconsistent - in some places the paper refers to 'Figure 1' but elsewhere there is 'figure 3' and 'figure 4' (on page 7). If references are to be capitalised, 'table 1' would also have to be replaced with 'Table 1'.

References:
[1] John Yang, Gyuejeong Lee, Simyung Chang, and Nojun Kwak. Towards governing agent’s efficacy: Action-conditional β-vae for deep transparent reinforcement learning. volume 101 of Proceedings of Machine Learning Research, pp. 32–47, Nagoya, Japan, 17–19 Nov 2019. PMLR. URL http: //proceedings.mlr.press/v101/yang19a.html.
[2]. Max Jaderberg, Volodymyr Mnih, Wojciech Marian Czarnecki, Tom Schaul, Joel Z Leibo, David Silver, and Koray Kavukcuoglu. Reinforcement learning with unsupervised auxiliary tasks. arXiv preprint arXiv:1611.05397, 2016.
[3] Joshua Achiam, Harrison Edwards, Dario Amodei, and Pieter Abbeel. Variational option discovery algorithms. arXiv preprint arXiv:1807.10299, 2018.

---

### Official Review · AnonReviewer1 · 2020-10-27
**A nice read, but not enough strong contributions.**

**Rating:** 3
**Confidence:** 5

**Review:**

First off, I would like to thank the authors of "Guiding Representation Learning in Deep Generative Models with Policy Gradients" for their neat, well written submission.


Summary of the paper
--------------------------------
The paper discusses the problem of representation learning in deep RL, with a focus on the effects of pretraing the state embedding function on agent performance. More specifically, the authors consider the case where a variational autoencoder (VAE) is to be trained to embed the state, and three training regimes are compared:
1. The VAE is pretrained on a dataset of episodes - collected separately with a pretrained agent. It weights are then frozen, and a policy is trained online.
2.  The VAE is pretrained on a dataset of episodes - collected separately with a pretrained agent. Its weights are then refined durign the training a policy online (gradients of the policy are backpropped into the VAE encoder).
3. The VAE and the policy are traiend end-to-end online jointly.

The authors' main claim is not only possible, but that pretraining without refinemenet / joint optimization can potentially result in the VAE discarding essential details needed for policy/value optimization; the claim is supported by:
1. Comparing the three training regimes on Breakout using a PPO;
2. Providing a visualization of the PPO value function highlighting how in the regimes in 2 and 3 details of the game that intuitively are relevant for good performance are indeed picked up during training.


Assessment
-----------------

-- The positives --

The paper is well structured and mostly well written - it was in fact a pleasure to read apart from a few confusing bits that I will point to later in the review.
The results are a bit thin in quantity, but to the point! The goal and the description of the experiments are clear and detailed, the figures clear and easy to understand.

-- The concerns --

1. The broad topic addressed in the paper (model training in RL) is interesting, but the contributions are not novel enough for this venue.

* First contrbution in Section 1: "we show that combined learning is possible and that it yields good performing policies"
Training models (variational or not) alongside a policy (in this or similar settings) is commonplace, and we now they can be trained. Example of much richer (action-conditional) and larger scale models than those presented in this submission are for example MuZero (https://arxiv.org/abs/1911.08265), or belief state models (one example https://arxiv.org/pdf/1906.09237.pdf).

* Second contribtion of Section 2: "we show that jointly trained representations incorporate additional, task-specific information which allows a RL agent to achieve higher rewards then if it was trained on a static representation"
This statement is quite unsurprising, as we are literally training the encoder to retain that information and provide it to the policy/value functions. Furthermore, the authors do not actually strictly verify that the fully pretrained model doesn't encode the ball location - that is, is the VAE decoder failing to reconstruct or the encoder failing to capture the ball localtion? What happens if you apply the method in Wang et al to PPO^{fixed}?

2. Whilst running on 1 environment with 1 training setup helps making the point, I can't help wondering if the comparisons between PPO^{VAE} and PPO^{adapt} could have yielded different results if multiple training tricks had been compared, e.g. something like using target networks for the VAE, using replay - which would have also made the comparison of the two method more fair to PPO^{VAE} since PPO^{adapt} has access to more compute.
The problem of how to use a pretrained model is indeed very exciting and I wish they authors had expanded more this section of the paper.

Suggestions
--

To make the paper more interesting, the authors could consider framing the paper in the broader context of unsupervised pretraining in DL (e.g. https://www.jmlr.org/papers/volume11/erhan10a/erhan10a.pdf) .

Please consider clarifying:

- Section 4.2 "This weight sharing method has proven to be more time efficient than to query the encoder n times
and concatenate the results afterwards." --> I find this a bit confusing - isn't the graph compiler going to optimized this out? Can you elaborate?

- Section 4.3. "While for PPOadapt, the latent space can be changed without restrictions, the decoder of PPOVAE constantly produces gradients do not contain information about the ball. " --> Sentence is badly formed, please fix.

- Section 4.3 "Therefore, PPOVAE has less incentive to achieve higher rewards, which is reflected by its performance." --> I do not see this, and would need to see an experiement to show/support this claim. Please point me to the relevant sections of the paper if I missed them.

- Conclusion "Our work verifies that the fundamental issue with pre-trained representations still exists and shows possible solutions in RL scenarios."  --> Why should the fundamental issue should have gone away? Also, the paper doesn't seem to suggest any solutions to improving the bridge between unsupervised pretraining and online training. Please do correct me if I am mistaken.


I look forward to the rebuttal, hoping the author will help me reassess the paper in a more positive light.

---

### Official Review · AnonReviewer4 · 2020-10-28
**Good general direction, but would like to see more thorough evaluation or novel formulation**

**Rating:** 4
**Confidence:** 4

**Review:**

This paper proposes a method for reinforcement learning with representations learned from a VAE. The VAE is used to encode the states (images) and the mean of the posterior is used as input to the policy. The VAE is trained using the variational lower bound and the policy is optimized using PPO. The model can be jointly trained with VAE and RL losses from scratch, or the VAE can be pre-trained and fixed, or pre-trained and finetuned.

Strengths:

+ This method is fairly simple and straightforward
+ The results suggest the representations learned can capture small details such as the ball which would normally not be reconstructed under the VAE loss
+ Pre-training the VAE seems to have promising results suggesting the pre-trained representations can improve learning

Concerns:
- This paper seems like a straightforward mash-up of VAE and PPO. It’s not clear whether the main contribution is particularly novel.
- No comparisons are made to non-VAE methods (e.g. a standard convolutional architecture). It’s not clear how much of the method’s performance can be attributed to the representation learning part vs just running a standard RL method from scratch
- Limited evaluation: there is one training curve on one Atari game, and there is one visualization of the jacobian. I would like to see a more thorough evaluation with comparisons to other baselines outside of this paper

Overall this paper is working on an interesting research direction, however I cannot recommend to accept it in its current state. I would like to see stronger empirical results or a more novel formulation.

---

### Official Review · AnonReviewer2 · 2020-10-31
**Poor literature review and poor experiments**

**Rating:** 1
**Confidence:** 5

**Review:**

The paper focused on the issue of learning a policy for a given task using the learned representations a pre-trained VAE. The authors visualize that using a learned latent space of a pre-trained VAE is not good enough for learning policies and propose a solution for this problem: back-propagate gradient policies through the VAE encoder. The authors proposed two versions on this method, one with pre-training and one fully online.

This paper suffers from fundamental issues. The biggest one in my opinion is its poor literature review. There is a bold statement in the introduction which shapes up the problem that the paper is trying to address and I quote: "Recently, many such approaches employed the VAE framework which aims to learn a smooth representation of its domain. Most of these approaches follow the same pattern: First, they build a dataset of states from the RL environment. Second, they train the VAE on this static dataset and lastly train the RL mode using the VAE’s representation." But the authors do not provide any reference for this. The related work section is also more focused on VAEs rather than its combination with RL. And as one would expect, there is no comparison with previous work either. Joint training of a VAE with a reward predictor or a policy network is not new. There is a massive body of research doing this. PlaNet, Dreamer by Danijar Hafner et al and their variations being the latest ones.

The experiments are also done on a single Atari game (Breakout) which is not sufficient to demonstrate the capability of the proposed method in a variety of tasks. And again, with no comparison to ANY of the previous work.

Overall, the paper is suffering from multiple fundamental issues which makes it hard to be accepted as a scientific contribution in a top conference.

---

### Decision · Program_Chairs · 2021-01-07
**Final Decision**

**Decision:**

Reject

**Comment:**

Due to uniformly unfavourable reviews and lack of author engagement in the discussion period, this paper is rejected.